# Neurofilament light chain as a potential biomarker for monitoring neurodegeneration in X-linked adrenoleukodystrophy

Isabelle Weinhofer[1], Paulus Rommer[2], Bettina Zierfuss [1], Patrick Altmann[2], Martha Foiani[3,4], Amanda Heslegrave[3,4], Henrik Zetterberg [3,4,5,6], Andreas Gleiss[7], Patricia L. Musolino[8], Yi Gong[8], Sonja Forss-Petter[1], Thomas Berger[2], Florian Eichler[8], Patrick Aubourg[9], Wolfgang Köhler[10] & Johannes Berger [1 ✉]

X-linked adrenoleukodystrophy (X-ALD), the most frequent monogenetic disorder of brain white matter, is highly variable, ranging from slowly progressive adrenomyeloneuropathy (AMN) to life-threatening inflammatory brain demyelination (CALD). In this study involving 94 X-ALD patients and 55 controls, we tested whether plasma/serum neurofilament light chain protein (NfL) constitutes an early distinguishing biomarker. In AMN, we found moderately elevated NfL with increased levels reflecting higher grading of myelopathy-related disability. Intriguingly, NfL was a significant predictor to discriminate non-converting AMN from cohorts later developing CALD. In CALD, markedly amplified NfL levels reflected brain lesion severity. In rare cases, atypically low NfL revealed a previously unrecognized smoldering CALD disease course with slowly progressive myelin destruction. Upon halt of brain demyelination by hematopoietic stem cell transplantation, NfL gradually normalized. Together, our study reveals that blood NfL reflects inflammatory activity and progression in CALD patients, thus constituting a potential surrogate biomarker that may facilitate clinical decisions and therapeutic development.

[1] Department of Pathobiology of the Nervous System, Center for Brain Research, Medical University of Vienna, Vienna, Austria. [2] Department of Neurology, Medical University of Vienna, Vienna, Austria. [3] UK Dementia Research Institute at UCL, London, UK. [4] Department of Neurodegenerative Disease, University College London, London, UK. [5] Department of Psychiatry and Neurochemistry, Institute of Neuroscience and Physiology, The Sahlgrenska Academy at the University of Gothenburg, Mölndal, Sweden. [6] Clinical Neurochemistry Laboratory, Sahlgrenska University Hospital, Mölndal, Sweden. [7] Section for Clinical Biometrics, Center for Medical Statistics, Informatics, and Intelligent Systems, Medical University of Vienna, Vienna, Austria. [8] Department of Neurology, Harvard Medical School, Massachusetts General Hospital, Boston, MA, USA. [9] Kremlin-Bicêtre Hospital, University Paris-Saclay, Paris, France. [10] Department of Neurology, Leukodystrophy Clinic, University of Leipzig Medical Center, Leipzig, Germany. ✉email: johannes.berger@meduniwien.ac.at

With a combined incidence of 1:14,700, the inherited neurodegenerative disorder X-linked adrenoleukodystrophy (X-ALD, OMIM #300100) is the most common peroxisomal disease[1]. The molecular cause of X-ALD are mutations in the *ATP-binding cassette subfamily D member 1 (ABCD1)* gene, which encodes a peroxisomal transporter mediating the import of coenzyme A-activated very-long-chain fatty acids (VLCFAs) into peroxisomes for degradation[2–4]. Accordingly, ABCD1 deficiency results in the accumulation of VLCFAs in tissues and body fluids of the patients[5]. Clinically, X-ALD patients show a striking phenotypic variability but no genotype–phenotype correlation exists[6]. The default manifestation of X-ALD that affects virtually all male X-ALD patients reaching adulthood and about 60% of female carriers is a dying-back axonopathy of sensory ascending and motor descending spinal cord tracts (adrenomyeloneuropathy, AMN) starting in young adulthood in males and later, with milder symptoms in heterozygous females[7]. AMN has a slow progression rate with changes determined by traditional and surrogate outcomes like the Expanded Disability Status Scale (EDSS), an accepted scoring system for leukodystrophies[8]. In addition to myeloneuropathy, around 80% of all male X-ALD patients develop adrenocortical insufficiency[9]. Whereas the adrenal insufficiency can be managed by steroid replacement therapy, no curative treatment is available for neurodegeneration in AMN. In more than half of all male X-ALD patients, with peak onset in boyhood, a still unknown trigger induces a severe rapidly progressive cerebral inflammatory demyelination (cerebral ALD, CALD) with the breakdown of the blood-brain barrier (BBB) and infiltration of mainly monocytes and T cells[10–12]. In the majority of cases, untreated CALD results in a vegetative state or death within a few years after disease onset[13]. If CALD is detected in the early stages before major neurological disability appears, the inflammatory demyelination can be arrested by hematopoietic stem cell transplantation (HSCT) or gene therapy[13–15]. Now that X-ALD has been added to newborn screening programs in several states of the U.S.A and in the Netherlands, a larger number of patients will benefit from early screening and lesion detection[16], enabling increased access and efficacy of presently available treatments.

Currently, detection of CALD relies on periodic screening by magnetic resonance imaging (MRI), where inflammatory cerebral white matter lesions with disruption of the BBB can be visualized by extravasation of gadolinium (Gd)-based contrast agents[17]. The clinical progression of CALD is graded using the MRI severity scoring of Loes, which applies a point system ranging from 0 to 34 based on both location and extent of cerebral demyelination as well as atrophy of the brain[18]. Although conventional MRI is able to detect the onset and progression of CALD, this method is associated with limitations as it does not directly report the degree of myelin loss and axonal injury. Thus, there is a pressing need for a sensitive biomarker directly reflecting neuronal integrity that could complement MRI analysis in X-ALD patients. So far, no biomarker is available to reveal the onset of CALD in its earliest stages, possibly before MRI scans visualize disturbance of the BBB. Such a biomarker, not confounded by BBB permeability, would be highly valuable to predict the expected neurological status and, therefore, quality of life of CALD patients after HSCT or gene therapy and could also support treatment decisions for patients with atypical brain lesion patterns, for whom the outcome of these interventions is uncertain.

The cytoskeletal protein neurofilament light chain (NfL) is the smallest of the three subunits that constitute neurofilaments, which are important scaffolding proteins in neurons[19]. Recently, NfL has been identified as a sensitive and reliable biomarker of neuroaxonal damage that indicates disease severity and/or progression in various neurodegenerative diseases including, for example, multiple sclerosis, amyotrophic lateral sclerosis, Alzheimer's, and Parkinson's disease[20–23]. In neurodegeneration and especially in inflammatory-related acute neuroaxonal damage, NfL from injured axons is released into the CSF and further into the blood where it can be measured using the ultrasensitive Single Molecule Array (SiMoA™) method[24]. Importantly, the concentrations of NfL in the blood are not affected by BBB permeability, which is key when applying a brain-specific biomarker for early and accurate diagnosis[25]. So far, neither NfL concentrations in CSF nor in blood have been reported for CALD patients.

The main objective of this study was to determine the potential of blood NfL as a prognostic marker for the onset and progression of CALD. Further, we assessed whether blood NfL levels would enable sensitive monitoring of activity and progression of the myeloneuropathy in AMN patients. We hypothesized that NfL concentrations in the blood: (a) are higher in individuals with asymptomatic X-ALD or AMN compared to age-matched healthy controls; (b) correlate with disease activity and progression, as assessed by EDSS score; (c) is further aggravated with the onset of inflammatory CALD with clinical progression graded by the MRI severity score, and (d) are lowered in CALD patients upon HSCT. Confirmation of these assumptions would establish NfL as a dynamic biomarker in the context of X-ALD and, thus, provide a complementing readout for the success of therapeutic interventions.

## Results

**Study design.** We included blood samples derived from 149 participants (94 X-ALD patients and 55 healthy volunteers) from four collaborating X-ALD centers. The final data set consisted of 61 AMN, 7 asymptomatic X-ALD, 24 CALD patients (5 patients from the AMN set that during the study converted to CALD were also included in the CALD set; during follow-up, two of the CALD patients received HSCT to halt the inflammatory demyelination), 5 CALD patients post-HSCT and 2 adolescent CALD patients in which the inflammatory demyelination spontaneously self-arrested. The control data set included blood samples from 48 healthy adults and 7 childhood/adolescent donors. Totaling 199 samples, our data set incorporated samples collected from 20 AMN patients at several time points during disease progression (53 longitudinal assessments) and from five CALD patients before and after conversion to inflammatory CALD (15 longitudinal assessments), as assessed by Gd-enhancement in MRI and clinical progression graded by the MRI severity score of Loes[18]. Clinical characteristics of the participants are shown in Table 1 and a more detailed description of the CALD patients in Supplementary Table 1. Based on previous investigations that demonstrated a high degree of correlation between plasma and serum NfL measurements[26] with slightly but in the univariable analysis not significantly elevated levels in serum and given the rarity of X-ALD, we decided to use both available plasma and serum to obtain a sample set of reasonable size. For all statistical analyses, we incorporated a sample type indicator for adjustment.

**NfL is mildly elevated in AMN and strongly amplified in CALD.** Concentrations of NfL in plasma and serum samples of asymptomatic X-ALD patients and controls did not differ significantly (Fig. 1a). In AMN patients with slowly progressive myeloneuropathy, NfL levels were moderately but significantly higher than in controls (raw data: 10.6 [8.0–14.9] vs. 5.7 [3.8–9.8] pg/ml; model estimate of mean ratio: 1.94, adjusted 95% confidence interval: 1.18–3.19; adj. $p = 0.004$); cf. Supplementary Fig. 1 for separate analyses of serum and plasma samples, Supplementary Fig. 2a for direct comparison with controls of similar age, and Supplementary Fig. 3 for statistical analysis without

**Table 1 Baseline group characteristics of X-ALD variants and healthy controls.**

| | Controls childhood/adolescent | Controls adult | X-ALD asymptomatic | X-ALD AMN |
|---|---|---|---|---|
| Participants, n | 7 | 48 | 7 | 61 |
| Samples, n | 7 | 49 | 8 | 93 |
| Age, years | 11 (10–12) | 39 (31–52) | 31 (26–40) | 40 (31–45) |
| EDSS | – | – | 0 (0–0) | 4.0 (3.5–6.0) |
| MRI severity Loes score | – | – | 0 (0–0) | 0 (0–0) |
| Blood NfL, pg/ml | 3.7 (3.6–5.3) | 6.5 (4.7–10.1) | 5.8 (4.4–7.2) | 10.6 (8.0–14.9) |

| | X-ALD CALD childhood/adolescent | X-ALD CALD adult | X-ALD CALD self-arrested | X-ALD CALD post HSCT |
|---|---|---|---|---|
| Participants, n | 13 | 11 | 2ᵃ | 7 |
| Samples, n | 19 | 13 | 2 | 8 |
| Age, years | 12 (9–15) | 44 (30–52) | 16 (16–16) | 26 (12–35) |
| EDSS | ND | 6 (3.5–6.5) | ND | ND |
| MRI severity Loes score | 11 (10–16) | 4 (3–10) | 1 (1–1) | 7 (3–13) |
| Duration post HSCT, years | – | – | – | 4.1 (1.0–6.7) |
| Blood NfL, pg/ml | 294.4 (142.1–923.9) | 55.8 (19.5–109.9) | 6.6 (5.7–7.5) | 21.5 (13.8–91.6) |

Continuous variables are median (IQR). CALD (cerebral ALD), childhood/adolescent-onset (age 3–21 years); CALD, adulthood onset (age >21 years), this group also includes five patients from the AMN set that throughout the study converted to CALD; CALD post-HSCT, CALD patients in which the cerebral demyelination was halted by hematopoietic stem cell transplantation (HSCT). The AMN group comprises patients with non-inflammatory myelopathy and X-ALD patients with myelopathy after spontaneously arrested (Gd-negative) CALD lesions.
*ND* not determined, *EDSS* Expanded Disability Status Scale.
ᵃTen additional patients with self-arrested CALD brain lesions and AMN symptoms are included in the AMN set.

longitudinal samples. With the onset of acute neuroinflammatory demyelination (CALD), NfL levels were markedly increased to levels by far exceeding those observed in AMN and controls (model estimate of mean ratio vs. AMN: 11.0; adj. 95% CI: 6.2–19.7; adj. $p < 0.001$; estimate of ratio vs. controls: 21.4; adj. 95% CI: 13.7–33.5; adj. $p < 0.001$; Fig. 1a). See Supplementary Fig. 2b for the display of the data and statistical analysis upon separation of the CALD group into childhood/adolescence and adulthood onset. The set of measurements at AMN status consisted of samples from patients with AMN that during follow-up did not convert to CALD ($n = 41$), as well as some that at the time of blood sampling were free of inflammatory involvement but later developed CALD (premanifest CALD, median interval until the diagnosis of conversion = 3.5 years post sampling, $n = 10$, a detailed description of the premanifest CALD patients is given in Supplementary Table 2), and patients with a known history of spontaneously arrested CALD episodes (CALD-arrested, $n = 10$). With the limitation of being based on a small sample set, we separately analyzed these AMN variants and observed significantly elevated NfL levels in the premanifest CALD group when compared to non-converting AMN patients (median 14.8 [10.3–16.7] vs. 9.4 [7.8–13.8] pg/ml, $p = 0.037$ Fig. 1b). To further corroborate these results, we performed a logistic regression analysis using ROC curves, in which the true-positive rate (sensitivity) is plotted against the false-positive rate with the area under the ROC curve (AUC) giving a measure of the accuracy of the test. Here, a value of 0.5 indicates a 50% probability of the test giving a correct answer, and 1 indicates a correct answer every time. This analysis indicated NfL as a significant predictor to discriminate samples from non-converting AMN vs. those from premanifest CALD (AUC 0.73). Aging (Fig. 1c) affected NfL in healthy controls (0.24 pg/ml increase per year) and asymptomatic X-ALD/AMN patients (0.14 pg/ml increase per year) to a similar extent ($p = 0.109$). In CALD patients presenting with highly elevated blood NfL, extensive axonal destruction masked age-related effects on blood NfL quantities (Fig. 1d).

**NfL strongly associates with MRI scored CALD brain lesion severity**. With the onset of CALD, blood NfL levels were markedly increased in the majority of affected X-ALD patients (Fig. 1a, d). To further understand the extent of NfL

amplification in individual CALD patients, we performed a linear regression analysis to assess the relationship between NfL and the activity of the inflammatory myelin destruction (Loes score of the MRI brain lesions). We found a statistically significant dependence of log(NfL) on the status of white matter abnormalities ($R^2 = 0.73$, $p = 0.002$), with more advanced CALD patients presenting both higher NfL and higher Loes scores (Fig. 2). There was no difference between the CALD cohort with childhood/adolescent-onset and patients developing CALD in adulthood. In CALD patients with very high Loes scores of ≥17, NfL levels were lower than in patients with Loes scores of 16, possibly due to more extensive advanced lesions with complete loss of large-caliber axons. In two boys and one adult patient with CALD identified at a very early disease stage (Loes MRI severity score of 1 and 2), NfL levels were still higher than in controls of similar age (Figs. 1d and 2). Longitudinal analysis in CALD patients revealed that with further lesional progression, NfL levels increased in seven out of eight patients. Correlation analysis of the slopes of NfL and Loes score changes over time showed a moderate positive association (Spearman's $r = 0.64$, $p = 0.086$; Supplementary Fig. 4).

**Low NfL despite marked brain lesion size reveals a previously unrecognized smoldering CALD course**. For three CALD patients (CALD1, CALD2, and CALD3), although rated with Loes scores ≥3, we found atypically low NfL levels (17.1, 11.1, and 22.0 pg/ml NfL, respectively, Fig. 2) when compared to other CALD patients with similar MRI brain lesion severity scores. Intriguingly, a detailed reassessment of clinical and MRI data of the disease courses in these CALD patients revealed that all three had an unusual, slowly progressing adulthood cerebral phenotype. Associated with the slow clinical worsening, we found slow lesion progression rates in MRI with hazy Gd-enhancement (smoldering lesions) that differed from the established rapidly progressive CALD disease course (Fig. 3).

**Onset of smoldering CALD is not immanent to marked NfL amplification**. To analyze whether the increase of NfL in the blood precedes leakage of the BBB indicative for the onset of acute inflammatory brain demyelination in both classical rapidly progressive CALD and smoldering CALD, we assessed NfL

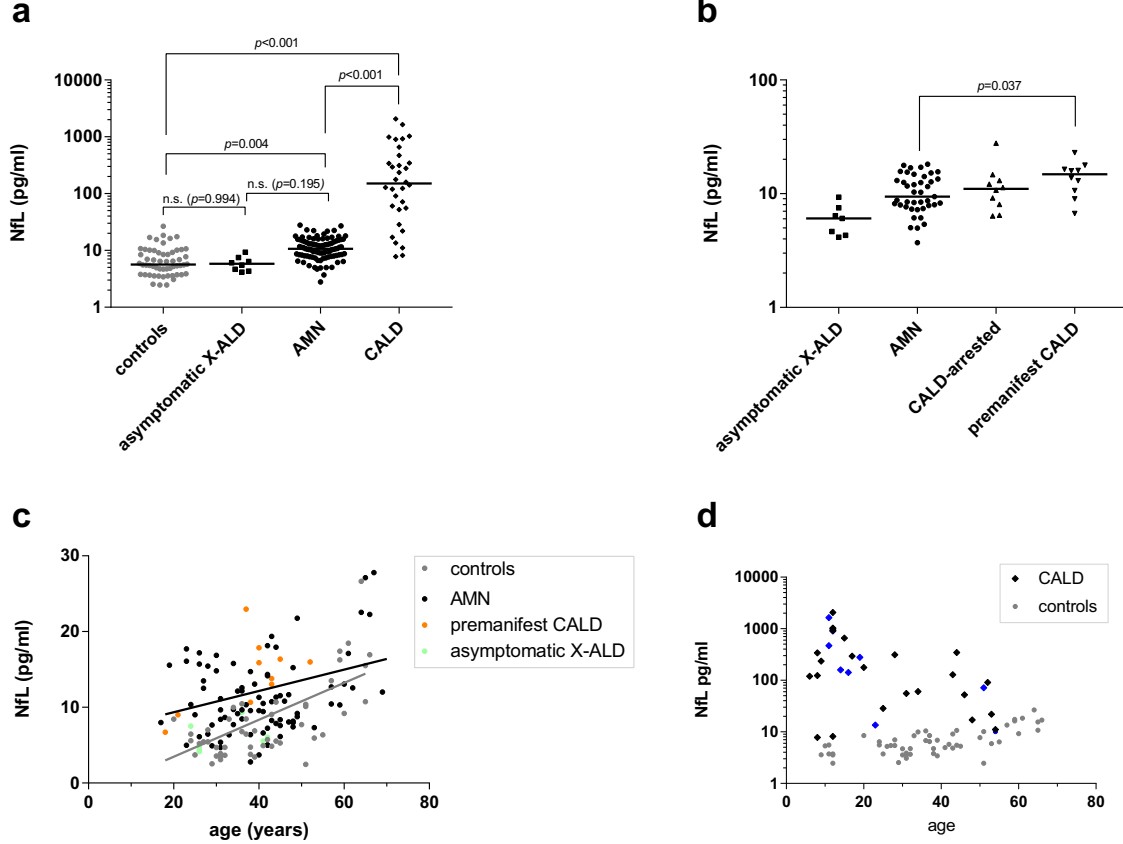

**Fig. 1 NfL levels in X-ALD patients. a** NfL in plasma and serum samples of asymptomatic X-ALD patients ($n = 7$, median age = 31 years, total sample number = 8), non-inflammatory AMN ($n = 61$, median age = 40 years, total sample number = 93), inflammatory CALD ($n = 24$, 13 childhood/adolescent CALD cases and 11 adult CALD cases, median age = 17 years, total sample number = 32), and healthy controls ($n = 55$, median age = 37 years, total sample number = 56). The median NfL level is indicated by a horizontal line. Total sample numbers include samples collected longitudinally from the same healthy control or patients during disease progression. Comparison of log(NfL) levels was done using a linear mixed model adjusted for sample type (serum vs. plasma) with the addition of a random ID factor to account for the longitudinal sampling of some individuals. Multiple testing was corrected by Tukey's method. Adjustment of the group comparisons for age differences did not change the significance or the reported *p*-values. **b** NfL in adult asymptomatic X-ALD patients ($n = 7$, mean age = 27 years) and patients with AMN status subdivided into AMN patients without brain involvement at the time of sampling (non-converting AMN, $n = 41$, median age = 42 years), AMN patients with self-arrested brain lesions (CALD-arrested, $n = 10$, median age = 39 years), and AMN patients that at the time of blood sampling were free of any inflammatory activity but later (median duration until diagnosed conversion = 3.5 years post sampling) developed CALD (premanifest CALD, $n = 10$, median age = 40 years). The median NfL level is indicated by a horizontal line. In cases of sampling at several time points during disease progression, the latest sample was used for display and analysis. The potential of NfL for discriminating AMN from premanifest CALD measurements is investigated in a logistic regression model with the group as a dependent variable and sample type as adjustment factor (two-sided test). **c** Linear regression of NfL on age in samples from X-ALD patients with asymptomatic (green data points) or AMN (black data points) status and adult healthy controls (gray data points). Data points in orange (premanifest CALD) indicate patients that were pre-symptomatic at baseline but converted to CALD later. **d** Association of NfL and age in samples from CALD patients (black and blue rhombuses) and healthy controls (gray filled circles). For longitudinal CALD samples, the latest time point is marked in black, and the preceding ones are indicated in blue. Source data are provided as a Source Data file.

longitudinally in five X-ALD patients before and after the appearance of Gd-enhancing lesions on the brain MRI. In X-ALD patients developing slowly progressing smoldering lesions (CALD1, CALD2), we found no change in NfL levels with the onset of CALD (Fig. 4a). In contrast, adult X-ALD patients who developed classical CALD with highly active lesions (CALD4, CALD8, and CALD9) demonstrated boosted levels of NfL associated with the presence of Gd-enhancement (Figs. 4b and 5).

**Halt of inflammatory demyelination by HSCT gradually normalizes NfL.** Next, we assessed how NfL responds to disease-modifying interventions and analyzed NfL in two CALD patients who underwent HSCT. By conducting longitudinal measurements of NfL prior to and up to 7 years after HSCT, we found

that although HSCT eventually stopped the neuronal damage, as reflected by decreased NfL levels, neuroaxonal destruction apparently continued up to 1-year post-HSCT (Fig. 5, case CALD9). This observation was further confirmed by assessment of NfL in five additional CALD patients with childhood/adolescent or adulthood onset, who had received HSCT, and by comparison to the NfL levels in CALD patients who experienced spontaneous arrest of cerebral demyelination (Supplementary Table 3).

**Higher prognostic value of blood NfL levels when compared with SDF-1.** We further investigated the association of NfL with CALD by comparing it to another blood biomarker indicative of neuroinflammation. Here, we used the chemokine stromal cell-

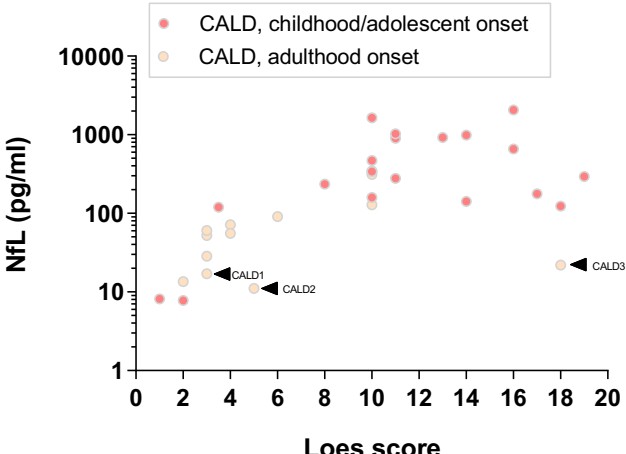

**Fig. 2 Association between plasma/serum NfL and MRI score of brain lesions in CALD patients.** The MRI score of location and activity of the inflammatory myelin destruction in CALD patients ($n = 24$, including 13 childhood/adolescent CALD cases [pink filled circles] and 11 adult CALD cases [beige filled circles], median age = 17 years, total sample number = 32), according to the 34-point severity scale of Loes, was associated with NfL levels ($R^2 = 0.73$, $p = 0.002$). Arrowheads indicate three patients with atypically mild and slowly progressive adult CALD disease course (CALD1-3). Statistical analysis included a fixed factor for sample type and a random ID factor as well as a linear and quadratic term for the Loes score. Source data are provided as a Source Data file.

derived factor 1 (SDF-1, also termed CXCL12) because it was previously shown to be elevated in the serum of boys with CALD[27] and also in patients with the neuroinflammatory demyelinating disorder multiple sclerosis[28,29]. Being a potent chemotactic factor, SDF-1 regulates the movement of monocytes and lymphocytes but also has a central role in neurogenesis. Using bead-array-based ELISA technology, we quantified SDF-1 in plasma and serum samples of CALD patients ($n = 20$) and healthy controls ($n = 23$, Supplementary Fig. 5). We found that SDF-1 was significantly increased in childhood/adolescent CALD patients ($n = 10$) when compared with healthy controls of similar age ($n = 7$, $p = 0.039$), thus confirming previous investigations (Supplementary Fig. 5a). However, SDF-1 did not differ significantly in the adult CALD sample set ($n = 10$) when compared with adult controls ($n = 16$, Supplementary Fig. 5a). Correlation analysis revealed no significant relationship between blood SDF-1 levels in samples derived from childhood/adolescent CALD patients and either MRI severity (Loes) scores (Spearman's $r = -0.52$, $p = 0.126$, Supplementary Fig. 5b) or blood NfL levels (Spearman's $r = -0.15$, $p = 0.693$; Supplementary Fig. 5c).

**Worsening of EDSS-graded myelopathy in AMN is reflected by increased NfL levels.** To elucidate whether NfL could be used as a quantitative outcome measure also for onset and progression of the non-inflammatory milder X-ALD variant, we compared the amount of NfL with the severity of the myelopathy in individual AMN patients as determined by the EDSS. We found that worsening of the myeloneuropathy in AMN patients is reflected by increased NfL levels, with each additional EDSS score point resulting in an average 6% increase in NfL (95% CI: 0.7–11.4%, $p = 0.026$), when adjusting for age at measurement and sample type (Fig. 6, cf. Supplementary Fig. 6 for the ex-post exclusion of AMN patients that later developed CALD, "premanifest CALD"). With EDSS being a neurological disability scoring system

originally developed for patients with multiple sclerosis, we additionally evaluated the neurologic dysfunction in AMN patients by correlating NfL levels to the motor function assessment based on the Adult Adrenoleukodystrophy Clinical Score (AACS). The AACS is a disease-specific scoring system that has been generated to differentiate clinical phenotypes in adult X-ALD[30]; details on the AACS gradings are shown in Supplementary Table 4. Whereas the analysis revealed a moderate correlation of NfL with AACS-graded motoric dysfunction in AMN patients (Spearman's $r = 0.376$), the average increase of 4.6% in NfL levels with each additional AACS-motor function grading point did not reach statistical significance ($p = 0.246$; Supplementary Fig. 7). NfL levels during the longitudinal progression of axonal degeneration in individual X-ALD patients with AMN status over a duration of up to 14 years are shown in Supplementary Fig. 8.

**Discussion**

In X-ALD, a disorder in which patients are affected by severe brain inflammation with rapid demyelination (CALD) and/or slowly progressive myelopathy (AMN), both resulting in axonal degeneration, we report exaggerated blood NfL with the onset of neuroinflammation. In a comparatively large set of X-ALD samples, we found that blood NfL is moderately but significantly elevated in AMN patients, reaching similar levels as observed in multiple sclerosis in remission (mean NfL, 17.0 pg/ml)[26], and strongly amplified with the onset of acute inflammatory brain demyelination in X-ALD patients, here aligning more with the levels reported in amyotrophic lateral sclerosis (median NfL, 125 pg/ml)[31]. A body fluid biomarker indicative for onset and progression of symptomatic X-ALD has not yet been established and the onset of life-threatening inflammatory CALD is assessed regularly by MRI analyses including the use of Gd-based contrast agents. In addition to concerns about Gd-accumulation in brain tissues upon frequent MRIs with Gd application[32], such analyses as outcome measure only reflect the focal breakdown of the BBB with variable degrees of myelin loss and axonal injury and thus, may be less quantitative than fluid biomarkers.

In the context of CALD, we observed that activity and progression of the inflammatory brain demyelination were highly associated with elevated NfL. In AMN patients who later developed inflammatory CALD (median duration between sampling and diagnosed conversion = 3.5 years), baseline NfL was significantly higher than in non-converting AMN patients. Although being an encouraging finding, some questions remain concerning the infrequent apparent absence of substantial axonal damage despite concurrent BBB disturbance observed in a small subset of CALD patients with atypical slowly progressive disease course that could complicate the adaption of NfL as an early-onset biomarker for imminent conversion to CALD. This group of adult patients diagnosed with ongoing CALD, all graded with Loes scores of ≥3, had only marginally elevated NfL when compared to the strikingly elevated levels in other CALD patients with similar MRI scores for brain lesions severity. In these patients, low NfL levels were indicative of an atypical, less aggressive CALD disease course with slow chronic smoldering inflammation and progression rates of neuronal damage. Thus, while pronounced Gd-enhancing CALD lesions clearly represent a strong inflammatory component with corresponding clinical symptoms, smoldering CALD plaques with hazy Gd-enhancement seem to be indicative of neuroinflammation sheltered behind an only partially disrupted BBB. Similar to patients with progressive multiple sclerosis, CALD patients with smoldering inflammation experience slow chronic deterioration over time[33]. It remains to

## rapidly progressive CALD (CALD4)

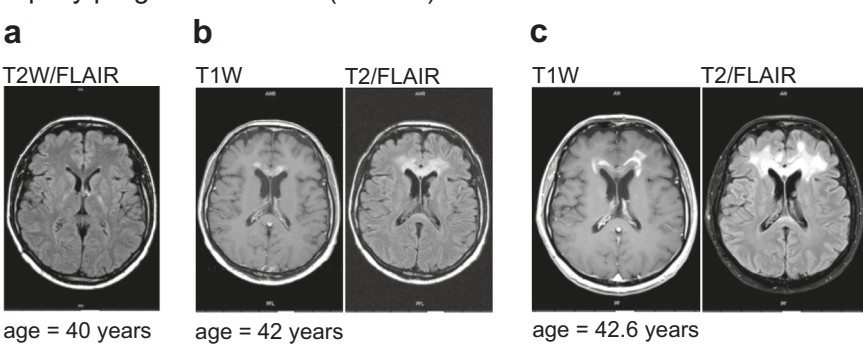

## slowly progressive, smoldering CALD (CALD2)

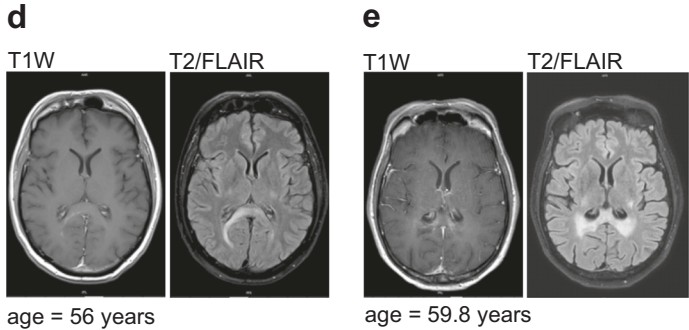

**Fig. 3 MRI analysis of slowly progressive smoldering inflammation in CALD. a** T2-weighted (FLAIR) sequence before the onset of CALD in patient CALD4 with classical, rapidly progressive disease course. **b, c** Frontal lobe cerebral inflammatory lesion with pronounced Gd-enhancement in T1-weighted (T1W) images and elevated FLAIR signal in the genu of the corpus callosum and frontal white matter, which was not seen two years earlier (**a**) and with the severe progression of the enhancing lesion seven months later. **d, e** MRI scans of a CALD patient (CALD2) with atypical, slowly progressive smoldering inflammation. **d** Hazy Gd-enhancement in T1W images and mildly elevated FLAIR signal in the splenium of the corpus callosum and posterior white matter. **e** Four years later, lesions had progressed mildly with diffuse marginal Gd-enhancement still notable, suggesting chronic smoldering inflammation.

be established whether, like in multiple sclerosis, the smoldering lesions in CALD expand due to sustained inflammatory processes driven by a rim of iron-laden microglia/macrophages at the lesion edge[34]. So why do some individuals - in our study adult CALD patients—develop chronic active lesions with relatively mild disease progression and limited BBB disruption and not the rapid disease course characteristic for classical CALD? Do these patients have altered susceptibility to CALD-associated inflammatory insults or a different repertoire of factors critical for progression? Did some of the rare CALD patients with spontaneously arrested lesion progression also initially present with such a smoldering disease course? Further research is needed to answer these questions. Currently, however, NfL as a biomarker directly reflecting the actual state of neuronal damage could be a tool to further differentiate classical CALD and atypical smoldering CALD disease courses. Thus, assessment of NfL in plasma or serum could complement current MRI-based strategies to score and monitor the disease status and thus, support decision making for treatment.

In CALD patients who had received HCST, we found that NfL gradually normalized in the blood approaching the levels observed in healthy controls. Although our data set is limited to few transplanted CALD patients, the NfL results are impressive and comparable to the correction of NfL observed after HSCT in particularly aggressive multiple sclerosis[35]. After HSCT of CALD patients, donor cell recovery reflected by blood neutrophil count is generally observed within 16 days and repair of the BBB within 100 days, as determined by lack of Gd-enhancement in MRI[36]. In

contrast, myelin destruction may continue up to one year after HSCT intervention. Although being a key determinant for neurologic disability, no information is available regarding the time required until the axonal destruction stops in these patients. Here, future prospective studies with more frequent longitudinal sampling prior to and after HSCT in increased CALD cohorts could provide detailed knowledge on how long axonal destruction continues after the halt of acute inflammatory demyelination, which would be valuable information for therapeutic interventions. So far, our results are highly encouraging for the utility of NfL levels as a surrogate marker to monitor axonal destruction in CALD patients during the evaluation of future strategies aimed to halt the inflammatory demyelination.

In AMN, the minor increase in NfL levels with EDSS-graded worsening of the myeloneuropathy and the moderate correlation with both EDSS- and AACS-graded motor functions (Spearman's $r = 0.354$ and Spearman's $r = 0.376$, respectively) are not entirely surprising, since AMN is characterized by a slow progression rate based on traditional outcome measures like EDSS. Thus, a lag in the evolution of disability after slow chronic axonal degeneration, with moderate NfL changes possibly preceding major functional decline until a critical threshold is reached, could account for the rather small additional increase in NfL with a higher EDSS rating of the AMN disease status. However, the shortcomings of the EDSS rating must be pointed out[37] with future studies involving possibly more sensitive outcome measures like diffusion-weighted MRI of the cervical spinal cord and the corticospinal tracts in the brain[38] having the potential to further clarify this issue. During

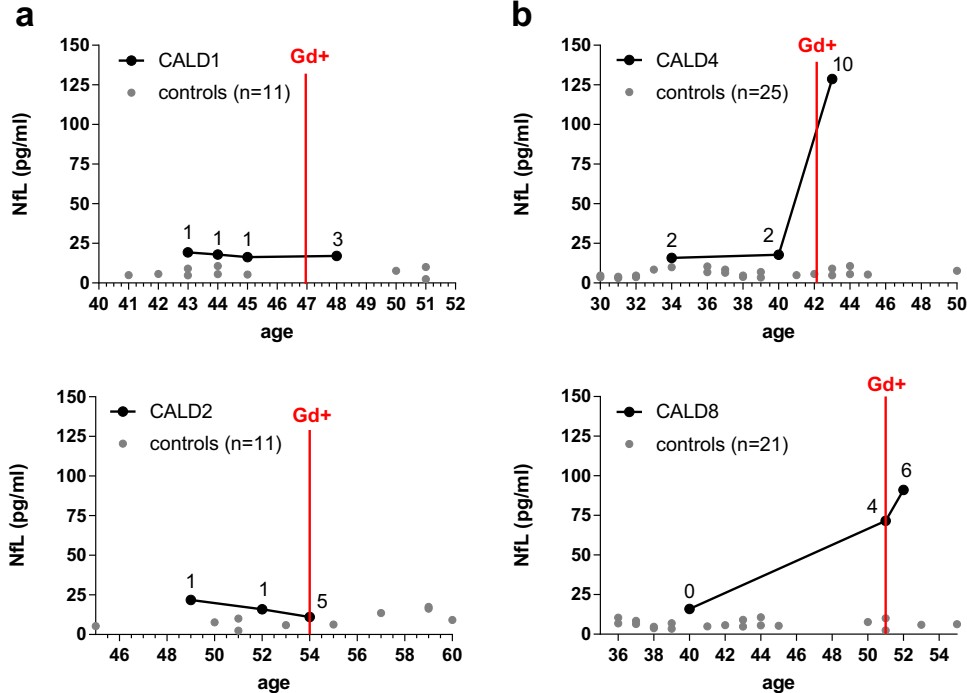

**Fig. 4 Longitudinal analysis of NfL in AMN patients before and after conversion to CALD. a** NfL levels up to five years before and after the onset of CALD, as indicated by the presence of Gd-enhancement (Gd+, vertical red line), in two patients developing an atypical, mild, and slowly progressive adult CALD disease course (CALD1 and CALD2). **b** NfL levels in two AMN patients, who developed classical, rapidly progressive CALD lesions (CALD4 and CALD8), 8–11 years before and with the start of Gd-enhancement. Numbers above each patient's data points indicate the corresponding MRI brain lesion severity Loes scores. For comparison, gray circles show NfL levels of the healthy control cohort in similar age ranges (**a**) age 41–51, $n = 11$; age 45-60, $n = 11$, (**b**) age 30–50, $n = 25$; age 36–55, $n = 21$. Source data are provided as a Source Data file.

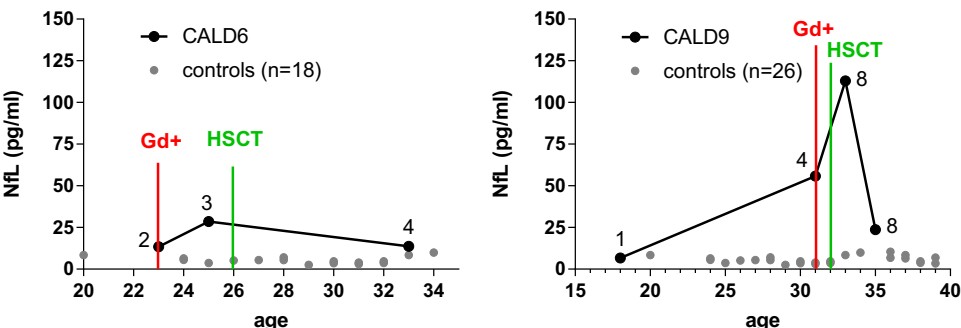

**Fig. 5 NfL levels in CALD patients before and after HSCT.** Longitudinal assessment of NfL in two adult CALD patients (CALD6 and CALD9) with classical CALD disease course before and after receiving HSCT. Numbers above each patient's data points indicate the corresponding Loes scores of brain lesions on MRI. For comparison, gray circles show NfL levels of the healthy control cohort in similar age ranges (age 20–34, $n = 18$; age 20–39, $n = 26$). The time point at which Gd-enhancement (Gd+) was initially detected is indicated by a vertical red line, the time point of HSCT intervention is marked by a green vertical line. Source data are provided as a Source Data file.

the revision of this manuscript, van Ballegoij and colleagues reported increased levels of NfL in both male and female AMN patients and found that in male patients, blood NfL was associated with clinical parameters of myelopathy as scored by EDSS, Severity Scoring system for Progressive Myelopathy (SSPROM) and timed up-and-go[39]. Accordingly, these investigations further support our observation that NfL may be of value also in the context of AMN.

Our study revealed substantial and important information on the plasma/serum levels of NfL as a measure of axonal destruction in the different disease courses of X-ALD. Nevertheless, some limitations should be acknowledged. Firstly, the

comparison between the stable AMN variants and AMN patients with premanifest CALD who later converted to acute CALD is based on a small sample size warranting some caution. This, together with our identification of an atypical smoldering CALD course characterized by the absence of the strong NfL amplification noted in classical CALD, currently limits the application of NfL as an early-onset marker for imminent conversion to CALD. Secondly, to understand in detail how the elevation of NfL in rapidly progressive CALD relates to the opening of the BBB and onset of inflammatory demyelination, more frequent longitudinal sampling is needed. Thirdly, for our findings to be generally applicable to the clinical management

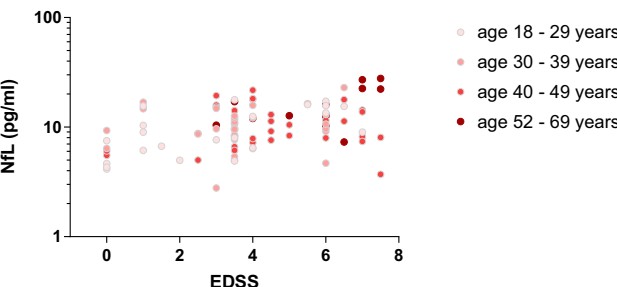

**Fig. 6 Worsening of EDSS-graded myelopathy in AMN is reflected by increased NfL.** The potential dependence of log of NfL on EDSS-graded clinical severity in individual asymptomatic X-ALD and AMN patients ($n = 68$, total sample number = 101 samples) was analyzed using a linear mixed model with fixed effects EDSS as well as age and sample type as adjustment variables and a random ID factor (two-sided test). The adjusted slope of the linear regression is back-transformed to the original scale, thus resulting in a percentage change, estimated as 6% higher NfL for each additional EDSS score point (95% CI: 0.7–11.4%, $p = 0.026$). Source data are provided as a Source Data file.

of CALD patients, with NfL levels supporting decision making and clinical interventions, a more detailed investigation of the predictive power of NfL in individual CALD patients will be needed with consideration also of a smoldering disease course and enlarged age-matched control groups for childhood/adolescent CALD patients. Finally, our study focused on NfL reflecting axonal damage and demonstrated the superiority of this biomarker in CALD in direct comparison to the blood levels of SDF-1, a chemotactic factor known to reflect inflammation and being elevated in children with CALD[27]. However, further studies are necessary to address and validate the prognostic power of NfL in CALD by opposing it to various other biomarkers proposed to indicate brain damage.

In summary, our study highlights the value of blood NfL as an easily accessible and quantifiable biomarker in the context of cerebral inflammatory X-ALD. With NfL levels directly reflecting the neuroaxonal damage, NfL measurements may support decision making for treatment especially in advanced CALD or in patients with atypical lesion patterns, where the efficacy of HSCT without major neurological deficits is uncertain. Furthermore, quantification of blood NfL levels should be a highly valuable readout to follow disease activity and progression in clinical trials targeting inflammatory demyelination and neurodegeneration in CALD patients.

## Methods
**Study design and participants.** This longitudinal multicenter cohort study was approved by the Ethical Review Board of the Medical University of Vienna (EK 1613/2019) and informed consent was obtained from participating X-ALD patients and healthy volunteers. All patients were male and diagnosed with X-ALD based on elevated plasma VLCFA by measurement of the total amount of the fatty acids C26:0, C24:0, C22:0 and, for normalization, C16:0 using gas chromatography-mass spectrometry (GC-MS). For the asymptomatic X-ALD ($n = 7$) and AMN ($n = 61$) cohorts, we included participants that presented without (asymptomatic X-ALD) or with symptoms of AMN according to EDSS, lacked acute inflammatory CALD brain lesions on MRI, had not received HSCT or gene therapy, had no other neurological disease interfering with the assessment of myelopathy, and had not participated in a clinical trial within the last year before blood sampling. CALD patients ($n = 24$) with Gd-enhancing brain MRI lesions at the time of blood sampling and clinical progression graded by the MRI severity score of Loes[18] were included if they had not undergone HSCT and had not been enrolled in clinical trials within the last year before blood sampling. The HSCT group ($n = 7$) consisted of participants that met the other inclusion criteria for CALD patients but received HSCT resulting in the halt of inflammatory brain lesions as established by the absence of Gd-enhancement on MRI. The childhood/adolescent CALD self-arrested set consisted of samples from two patients in which CALD inflammatory brain lesions spontaneously self-arrested in childhood/adolescence and that were asymptomatic for AMN symptoms. The healthy control cohort consisted of 48 adult and 7 childhood/adolescent male participants with no diagnosis of disorders characterized by neuroinflammation and/or axonal degeneration, such as X-ALD, amyotrophic lateral sclerosis, multiple sclerosis, or any dementia.

**Blood sampling and NfL measurement.** Blood samples from X-ALD patients and healthy controls were collected between 1992 and 2019 at four collaborating X-ALD research sites (University of Leipzig Medical Center, Germany; Massachusetts General Hospital, U.S.A.; University Paris-Saclary, Kremlin-Bicêtre hospital, France; Medical University of Vienna, Austria) into standard polypropylene EDTA test tubes and processed at room temperature within 2 h. For serum, tubes were left upside down at room temperature for 30 min to allow clotting before centrifugation at 2000×$g$ for 15 min at room temperature. For plasma, the blood samples were promptly centrifuged at 3500×$g$ for 15 min. The obtained serum and plasma samples were aliquoted, pseudonymized, and stored at −80 °C.

NfL in plasma/serum is stable at −80 °C and insensitive to repeated freeze-thawing. We accepted both serum and plasma samples, as NfL levels show a high degree of correlation between these two sample types[26]. NfL levels were assessed for blinded samples using SiMoA™ technology on an HD-1 Analyzer, according to the manufacturer's instructions (Quanterix, Billerica, MA). Briefly, serum/plasma samples were thawed, vortexed, and centrifuged at 10,000×$g$ for 5 min at room temperature. On-board the instrument, 1:4 diluted samples were bound to paramagnetic beads coated with a capture antibody specific for human NfL. The NfL-bound beads were incubated with a biotinylated NfL detection antibody followed by streptavidin-β-galactosidase enzyme complex and subsequent hydrolysis reaction with resorufin β-D-galactopyranoside substrate producing a fluorescent signal proportional to the concentration of captured NfL. Samples were measured in duplicates and NfL concentrations (pg/ml) were extrapolated from a standard curve, fitted using a 4-parameter logistic algorithm. Intra and inter-assay coefficients of variation were determined and found to be <20%.

With the exception of the seven childhood control samples, all samples were measured at once at a single institution.

**SDF-1 measurement.** The chemokine SDF-1 was measured in plasma and serum samples using Luminex® technology and the Human Cytokine/Chemokine Panel II (Merck) according to the manufacturer's instruction.

**Statistical analysis.** The distribution of NfL is described using median and quartiles due to a right-skewed distribution. Comparisons of the log of NfL between groups of measurements (controls, asymptomatic X-ALD, AMN, CALD) is done using a linear mixed model that adjusts the group factor for sample type (serum vs. plasma) as a fixed factor. A random ID factor is added to take replications within patients into account. Post-hoc pair-wise group comparisons are corrected for multiple testing using Tukey's method. Pair-wise group differences are back-transformed to ratios of geometric means (with adjusted 95% confidence intervals). The potential of NfL for discriminating AMN from premanifest CALD measurements is investigated using a logistic regression model with a group as a dependent variable and sample type as an adjustment factor. The area under the receiver operating characteristic curve (AUC) is reported as a measure of discrimination. Age dependence of NfL is investigated using a linear mixed model with indicators for the group of measurements and for the sample type, age as a covariable as well as the interaction of age and group, and a random ID factor. The difference of age slopes is tested via the interaction term. The dependence of log of NfL on the Loes score includes, besides a sample type fixed factor and a random ID factor, a linear and a quadratic term for Loes. The total influence of Loes (linear and quadratic term) is tested using an $F$-test with 2 degrees of freedom.

The potential influence of sample type (serum vs. plasma) on the difference of NfL between control and AMN measurements is investigated using a mixed model with fixed effects for measurement group (control vs. AMN) and for sample type as well as their interaction and a random ID factor. The potential dependence of log of NfL on EDSS (or AACS) within AMN measurements are analyzed using a linear mixed model with fixed effects EDSS (or AACS) as well as age and sample type as adjustment variables, and a random ID factor. For a descriptive investigation of a potential association of changes over time (NfL with Loes), the repeated measurements for each individual are regressed on time. The resulting individual slopes are then used as variables for a Spearman correlation coefficient. Correlation coefficients between −0.3 and +0.3 are regarded to indicate no association.

$p$-values below 0.05 are regarded to indicate statistical significance. All graphs were produced using GraphPad Prism 7.00. All statistical results, including $p$-values and regression lines within graphs, are calculated using SAS 9.4. Based on preliminary experiments, the sample size has been calculated using the nQuery software (version 8). For data collection, Microsoft Excel 2016 and IBM SPSS Statistics Version 21 were used.

**Reporting summary**. Further information on research design is available in the Nature Research Reporting Summary linked to this article.

## Data availability

All data are included in the manuscript or in the Supplementary Materials and are available from the corresponding author upon request. Source data are provided with this paper.

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

## Acknowledgements

We would like to thank all patients and healthy volunteers who participated in this study. This work was supported by the Austrian Science Fund KLI 837-B and T562-B13 to I.W. and DK W1205 and P26112-B19 to J.B. as well as by the European Leukodystrpophy Association (ELA) Germany and ELA International 2018-003I2 and ELA 2020-003C1A to J.B. B.Z. is supported by the Austrian Science Fund DK W1205. H.Z. is a Wallenberg Scholar supported by grants from the Swedish Research Council (#2018-02532), the European Research Council (#681712), Swedish State Support for Clinical Research (#ALFGBG-720931) and the UK Dementia Research Institute at UCL. The HD-1 Analyzer was purchased using a Wellcome Trust Multi-User Equipment Grant to H.Z. and A.H. The funders of the study had no role in the design, data collection, data analyses, data interpretation, or writing of the article. The corresponding author had full access to all the data and had final responsibility for the decision to submit the study for publication.

## Author contributions

I.W., J.B., and P.R. designed the study. Study conception, data acquisition, and analysis were contributed by I.W., J.B., W.K., P.R., and S.F.-P. H.Z., M.F., and A.H. conducted the NfL measurements and A.G. the statistical analysis. W.K., P.Au., F.E., B.Z., and T.B. contributed to patient recruitment, collection, and interpretation of the clinical data. P.A., P.L.M., and Y.G. contributed to recruiting patients and sample collection. I.W. created the figures and wrote, J.B. and S.F.P. edited the manuscript. All authors critically reviewed the manuscript and approved the final draft.

## Competing interests

H.Z. has served at scientific advisory boards for Denali, Roche Diagnostics, Wave, Samumed, and CogRx, has given lectures in symposia sponsored by Fujirebio, Alzcure, and Biogen, and is a co-founder of Brain Biomarker Solutions in Gothenburg AB, a GU Ventures-based platform company at the University of Gothenburg (all unrelated to the submitted work). The remaining authors declare no competing interests.

**Additional information**

