## [Peer Review File · Nature Communications]

Reviewer #1 (Remarks to the Author):

Title: Neurofilament light chain as a biomarker for monitoring neurodegeneration in X-linked adrenoleukodystrophy

Main message: The authors samples blood from 94 patients with X-ALD and 48 controls and measured levels of NfL. They found that in the slowly progressing AMN, NfL was moderately elevated – and elevations correlated to myelopathy related disability. NfL was a predictor of patients later developing CALD (vs non-converting AMN). In CALD, NfL levels were associated with brain lesion severity. Following hematopoietic stem cell transplantation, NfL levels normalized, thus could act as a surrogate marker of treatment efficacy.

English language: Adequate throughout

Statistical approach: Adequate methods used on the data available. Age and serum/plasma compartment well adjusted for with current methods.

Main issues:

1. Why did you only measure NfL? I agree that it shows a lot of promise, but there could be several other markers of disease that shows similar trajectories and trends in this cohort as there are a multitude of other analytes suggested to be good biomarkers. As no other marker is studied, it is difficult to say if NfL is superior to them in any way. This should be elaborated on as a limitation.
2. While you do try to correlate NfL levels to lesion severity on MRI, you never correlate it to inflammatory markers or laboratory markers of BBB disruption (e.g. albumin quota in serum:csf). I believe this should be highlighted as a potential limitation as it would have further supported your claim of NfL as a marker of BBB and increased inflammatory activity.
3. The specificity (and sensitivity) of NfL in these conditions is poor. A lot of patients with CALD had same NfL levels as healthy age matched controls, and while significantly different the AMN group had a huge overlap of NfL levels to healthy controls. This is elaborated poorly on in the Discussion section. Furthermore, while I understand that reference levels and cut-offs are to premature to suggest, I believe that the levels seen in this paper should be elaborated more on in comparison to other diseases (you briefly mention MS).

Specific issues:

4. Figure 3: While the FLAIR is a T2W signal I think it should be mentioned that Fig 3 shows FLAIR sequences.
5. Please use only 3 significant numbers (e.g. $p < 0.001$ and not 0.0012). Your use of 3 or 4 varies throughout the manuscript.

Reviewer #2 (Remarks to the Author):

This is a well designed and intriguing study assessing the suitability of NfL as a quantitative prognostic marker of disease severity and progression in X-ALD. It shows for the first time that X-ALD patients (both AMN and CALD children and adults) exhibit higher levels of NfL in blood with a more marked increase in inflammatory CALD patients. In addition, the results indicate that AMN patients who converted to inflammatory CALD, showed a higher quantity of NfL prior to conversion. Also, NfL levels were found to normalize in inflammatory CALD patients who received HCST, which is an important finding. The authors present NfL as a novel biomarker of severity and progression for X-ALD, able to predict conversion to CALD and recovery after HCST. The authors discuss the NfL may hold potential to inform decisions together with gadolinium enhancing, and be useful to monitor efficacy of treatments. This is a relevant study for the X-ALD field, which adds to

the body of evidence on NfL as a biomarker for neurodegenerative diseases.

I would like to express some concerns that may require major modifications:

1-In my view, the EDSS is a good qualitative scale to assess the general status of patients with AMN, however I think that it is not a scale sensitive enough to be able to detect small changes in patients. It would therefore be advisable to correlate NfL levels to more quantitative scales such as 6mWT or 2mWT in these patients as an outcome measure, or to the MRI lesions in cervical spinal cord as discussed; perhaps a more in-depth discussion on the limitations of EDSS is required here.

2-Is not clear to this reviewer why there is no control samples for the CALD children; the levels of NfL should be quantified also in control children.

3-The legends of Figure 1, Figure 2, and Figure 6 do not include the statistical analysis used, needs to be corrected.

4-In Figure 1A, the child/adolescent and adult CALD data are put together; I would very much prefer to see the results when the two groups are separated. Also, I do not understand why several samples of the same patient are used. In AMN, there is n=61 patients, but 93 samples; in asymptomatic X-ALD there are n=7 patients but 8 samples; in CALD there are n=24 patients but 32 samples; and there are n=48 healthy individuals but 49 samples...Are those replicates of the same sample? How would the analysis look like without replications from the same sample?

5-In Figure 1B, AMN patients are divided into AMN, CALD arrested, and premanifest CALD. Why did the authors use only one replicate here? The mean that we are seeing in the figure, is the mean of the replicate?

6-Figure 4B and 5 shows CALD4, CALD8, and CALD9 patients who developed classical CALD with highly active lesions and exhibited boosted levels of NfL associated with the presence of Gd-enhancement. The authors have plasma of these 3 patients with the two different phenotypes (AMN and CALD). Have the authors tried to do a statistical analysis of NfL levels after conversion of AMN into CALD?

7-In Figure 6, the authors show that levels of NfL correlate to an increase of EDSS, chosen as a relevant outcome measure. They used all the replicates (101 samples but n=68 patients) from asymptomatic X-ALD and from AMN patients, as in Figure 1A. In Figure 1B, the authors observe that levels of NfL were significantly higher in AMN patients who converted to CALD. What would the results be regarding correlation to EDSS if these samples (AMN patients who developed classical CALD) are removed for the analysis? Would the correlation between EDSS and NfL levels hold true?

POINT-BY-POINT RESPONSE Weinhofer et al. “Neurofilament light chain as a biomarker for monitoring neurodegeneration in X-linked adrenoleukodystrophy”

REVIEWER COMMENTS

Reviewer #1 (Remarks to the Author):

Title: Neurofilament light chain as a biomarker for monitoring neurodegeneration in X-linked adrenoleukodystrophy

Main message: The authors samples blood from 94 patients with X-ALD and 48 controls and measured levels of NfL. They found that in the slowly progressing AMN, NfL was moderately elevated – and elevations correlated to myelopathy related disability. NfL was a predictor of patients later developing CALD (vs non-converting AMN). In CALD, NfL levels were associated with brain lesion severity. Following hematopoietic stem cell transplantation, NfL levels normalized, thus could act as a surrogate marker of treatment efficacy.

English language: Adequate throughout

Statistical approach: Adequate methods used on the data available. Age and serum/plasma compartment well adjusted for with current methods.

Main issues:

1. Why did you only measure NfL? I agree that it shows a lot of promise, but there could be several other markers of disease that shows similar trajectories and trends in this cohort as there are a multitude of other analytes suggested to be good biomarkers. As no other marker is studied, it is difficult to say if NfL is superior to them in any way. This should be elaborated on as a limitation.
2. While you do try to correlate NfL levels to lesion severity on MRI, you never correlate it to inflammatory markers or laboratory markers of BBB disruption (e.g. albumin quota in serum:csf). I believe this should be highlighted as a potential limitation as it would have further supported your claim of NfL as a marker of BBB and increased inflammatory activity.
3. The specificity (and sensitivity) of NfL in these conditions is poor. A lot of patients with CALD had same NfL levels as healthy age matched controls, and while significantly different the AMN group had a huge overlap of NfL levels to healthy controls. This is elaborated poorly on in the Discussion section. Furthermore, while I understand that reference levels and cut-offs are premature to suggest, I believe that the levels seen in this paper should be elaborated more on in comparison to other diseases (you briefly mention MS).

Specific issues:

4. Figure 3: While the FLAIR is a T2W signal I think it should be mentioned that Fig 3 shows FLAIR sequences.

5. Please use only 3 significant numbers (e.g. $p < 0.001$ and not 0.0012). Your use of 3 or 4 varies throughout the manuscript.

Reviewer #2 (Remarks to the Author):

This is a well designed and intriguing study assessing the suitability of NfL as a quantitative prognostic marker of disease severity and progression in X-ALD. It shows for the first time that X-ALD patients (both AMN and CALD children and adults) exhibit higher levels of NfL in blood with an more marked increase in inflammatory CALD patients. In addition, the results indicate that AMN patients who converted to inflammatory CALD, showed a higher quantity of NfL prior to conversion. Also, NfL levels were found to normalize in inflammatory CALD patients who received HCST, which is an important finding. The authors present NfL as a novel biomarker of severity and progression for X-ALD, able to predict conversion to CALD and recovery after HCST. The authors discuss the NfL may hold potential to inform decisions together with gadolinium enhancing, and be useful to monitor efficacy of treatments. This a relevant study for the X-ALD field, which adds to the body of evidence on NfL as a biomarker for neurodegenerative diseases.

I would like to express some concerns that may require major modifications:

1-In my view, the EDSS is a good qualitative scale to assess the general status of patients with AMN, however I think that it is not a scale sensitive enough to be able to detect small changes in patients. It would therefore be advisable to correlate NfL levels to more quantitative scales such as 6mWT or 2mWT in these patients as an outcome measure, or to the MRI lesions in cervical spinal cord as discussed; perhaps a more in-depth discussion on the limitations of EDSS is required here.

2-Is not clear to this reviewer why there is no control samples for the CALD children; the levels of NfL should be quantified also in control children.

3-The legends of Figure 1, Figure 2, and Figure 6 do not include the statistical analysis used, needs to be corrected.

4-In Figure 1A, the child/adolescent and adult CALD data are put together; I would very much prefer to see the results when the two groups are separated. Also, I do not understand why several samples of the same patient are used. In AMN, there is n=61 patients, but 93 samples; in asymptomatic X-ALD there are n=7 patients but 8 samples; in CALD there are n=24 patients but 32 samples; and there are n=48 healthy individuals but 49 samples...Are

those replicates of the same sample? How would the analysis look like without replications from the same sample?

5-In Figure 1B, AMN patients are divided into AMN, CALD arrested, and premanifest CALD. Why did the authors use only one replicate here? The mean that we are seeing in the figure, is the mean of the replicate?

6-Figure 4B and 5 shows CALD4, CALD8, and CALD9 patients who developed classical CALD with highly active lesions and exhibited boosted levels of NfL associated with the presence of Gd-enhancement. The authors have plasma of these 3 patients with the two different phenotypes (AMN and CALD). Have the authors tried to do a statistical analysis of NfL levels after conversion of AMN into CALD?

7-In Figure 6, the authors show that levels of NfL correlate to an increase of EDSS, chosen as a relevant outcome measure. They used all the replicates (101 samples but n=68 patients) from asymptomatic X-ALD and from AMN patients, as in Figure 1A. In Figure 1B, the authors observe that levels of NfL were significantly higher in AMN patients who converted to CALD. What would the results be regarding correlation to EDSS if these samples (AMN patients who developed classical CALD) are removed for the analysis? Would the correlation between EDSS and NfL levels hold true?

Specific comments to referee 1:

1. Why did you only measure NfL? I agree that it shows a lot of promise, but there could be several other markers of disease that shows similar trajectories and trends in this cohort as there are a multitude of other analytes suggested to be good biomarkers. As no other marker is studied, it is difficult to say if NfL is superior to them in any way. This should be elaborated on as a limitation.

We thank the reviewer for this valuable suggestion and agree that the comparison of NfL to another biomarker would more precisely indicate the prognostic value of NfL. Accordingly, we decided to measure the chemokine SDF-1/CXCL12, a neuroinflammatory marker, in our CALD plasma/serum sample set, because this marker has been associated with demyelinating neuroinflammatory disorders like multiple sclerosis (Azin et al., J Mol Neurosci 2012; Khorramdelazad et al., J Neuroimm 2016) and was found elevated in a study involving 36 boys with CALD (Lund et al., Plos One 2012). Accordingly, we performed a Luminex bead array-based ELISA to determine SDF-1 levels in our CALD and control sample set and added this data as new Suppl. Fig 5. We found SDF-1 to be marginally but significantly elevated in samples derived from childhood/adolescent CALD patients when compared to age-matched controls, thus confirming the finding by Lund et al. However, neither Loes score of CALD brain lesion severity nor NfL levels correlated with SDF-1 in our sample set. In addition, we were unable to detect a significant increase of SDF-1 in adult CALD patients. Together, these results demonstrate that with NfL being highly elevated in both childhood/adolescent and adult CALD

patients and correlating with brain lesion severity, this biomarker is by far superior to blood SDF-1. Clearly, we cannot rule out that other inflammatory markers, not yet tested in the context of CALD, or possibly a combinatorial set of such markers might even better reflect onset and progression of CALD. We have added this statement as a limitation to the discussion section.

Changes in the manuscript:

Results, p.8

Higher prognostic value of blood NfL levels when compared with SDF-1

We further investigated the association of NfL with CALD by comparing it to another blood biomarker indicative for neuroinflammation. Here, we used the chemokine stromal cell-derived factor 1 (SDF-1, also termed CXCL12) because it was previously shown to be elevated in the serum of boys with CALD²⁶ and also in patients with the neuroinflammatory demyelinating disorder multiple sclerosis^{27,28}. Being a potent chemotactic factor, SDF-1 regulates the movement of monocytes and lymphocytes but also has a central role in neurogenesis. Using bead-array based ELISA technology, we quantified SDF-1 in plasma and serum samples of CALD patients (n=20) and healthy controls (n=22, Fig. S5). We found that SDF-1 was significantly increased in childhood/adolescent CALD patients (n=10) when compared with healthy controls of similar age (n=7, p=0.039), thus confirming previous investigations (Fig. S5a). However, SDF-1 did not differ significantly in the adult CALD sample set (n=10) when compared with adult controls (n=16, Fig. S5a). Correlation analysis revealed no significant relationship between blood SDF-1 levels in samples derived from childhood/adolescent CALD patients and either MRI severity (Loes) scores (r Spearman=-0.52, p=0.126, Fig. S5b) or blood NfL levels (r Spearman=-0.15, p=0.693; Fig. S5c).

Fig. S5. No correlation between blood SDF-1 and NfL levels in CALD patients. a SDF-1 levels in childhood/adolescent CALD (n=10), adult CALD (n=10), childhood/adolescent controls (n=7) and adult controls

(n=16). Statistical analysis was carried out using a two-ANOVA model with interaction. b Correlation analysis between SDF-1 levels and Loes score (r Spearman=-0.52; p=0.126). c Correlation analysis between SDF-1 levels and NfL (r Spearman=-0.15 partialized for sample type (serum vs. plasma); p=0.693).

Discussion, p.11

Finally, our study focused on NfL reflecting axonal damage and demonstrated the superiority of this biomarker in CALD in direct comparison to the blood levels of SDF-1, a chemotactic factor known to reflect inflammation and being elevated in children with CALD ²⁶. However, further studies are necessary to address and validate the prognostic power of NfL in CALD by opposing it to various other novel biomarkers proposed to indicate brain damage.

Materials and Methods, p. 13

SDF-1 measurement

The chemokine SDF-1 was measured in plasma and serum samples using Luminex® technology and the Human Cytokine/Chemokine Panel II (Merck) according to manufacturer's instruction.

2. While you do try to correlate NfL levels to lesion severity on MRI, you never correlate it to inflammatory markers or laboratory markers of BBB disruption (e.g. albumin quota in serum:csf). I believe this should be highlighted as a potential limitation as it would have further supported your claim of NfL as a marker of BBB and increased inflammatory activity.

For NfL it was recently reported that its concentration in the blood is not confounded by BBB permeability, as shown by the lack of correlation between serum NfL and CSF/serum albumin ratio (Kalm et al., Brain Res. 2017). We do believe that this aspect is of major importance for our study aimed to identify a brain-specific marker for early and accurate diagnosis, possibly indicative even before BBB disruption is detectable by MRI studies involving Gd-enhancement or altered CSF/serum albumin ratio. In order to avoid confusion and to further clarify this important aspect in our manuscript, we have included the following changes:

Changes in the manuscript:

Introduction, p.4

Such a biomarker, not confounded by BBB permeability, would be highly valuable to predict the expected neurological status and, therefore, quality of life of CALD patients after HSCT or gene therapy and could also support treatment decisions for patients with atypical brain lesion patterns, for whom the outcome of these interventions is uncertain.

Importantly, the concentrations of NfL in the blood are not affected by BBB permeability, which is key when applying a brain-specific biomarker for early and accurate diagnosis ²⁴.

3. The specificity (and sensitivity) of NfL in these conditions is poor. A lot of patients with CALD had same NfL levels as healthy age matched controls, and while significantly different

the AMN group had a huge overlap of NfL levels to healthy controls. This is elaborated poorly on in the Discussion section.

We strongly agree with the reviewer that the application of a biomarker requires high specificity and sensitivity. We further agree that solely considering Fig. 1a could leave the impression that a significant proportion of CALD patients overlap with the control sample set. However, as plasma/serum NfL is known to rise with age, we have extended our control data set with samples derived from healthy control children, as also suggested by reviewer 2. In new Fig. 1D, we used this enlarged control group to directly compare NfL levels and age in both CALD patients and controls. In combination with Fig. 2 and Fig. 3, the data show that in total 5 out of 24 CALD patients (total sample number including longitudinal samples taken at different time points during disease progression = 7) seem to be close to or within the range of healthy controls: Two of these five patients are children that were diagnosed with CALD at a very early disease stage (Loes MRI severity score of 1 and 2). In both boys, NfL levels were elevated when compared to age-matched controls (7.8 and 8.2 pg/ml vs. mean 3.99 and median 3.72 pg/ml of seven controls). For three adult CALD patients, NfL levels overlapped with the control group and, interestingly, in all cases their medical records revealed an atypical smoldering CALD disease course (Fig. 3).

Future studies using an enlarged sample set derived from healthy control children and from CALD boys very early after onset are required for detailed comparisons and statistical analyses of childhood/adolescent CALD and age-matched controls. In addition, we realise that the rare cases of smoldering CALD currently may obscure the specificity and sensitivity of NfL in the context of X-ALD. Accordingly, we have further outlined these two limitations in the Discussion section.

Changes in the manuscript:

Results, p.5

The control dataset included blood samples from 48 healthy adult and from 7 childhood/adolescent donors. Totalling 199 samples, our dataset incorporated samples collected from 20 AMN patients at several time points during disease progression (53 longitudinal assessments) and from five CALD patients before and after conversion to inflammatory CALD (15 longitudinal assessments), as assessed by Gd-enhancement in MRI and clinical progression graded by the MRI severity score of Loes¹⁷.

NfL is mildly elevated in AMN and strongly amplified in CALD

In AMN patients with slowly progressive myeloneuropathy, NfL levels were moderately but significantly higher than in controls (raw data: 10.6 [8.0–14.9] vs. 5.7 [3.8–9.8] pg/ml; model estimate of mean ratio: 1.94, adjusted 95% confidence interval: 1.18 – 3.19; adj. p=0.004); cf. Fig. S1 for separate analyses of serum and plasma samples, Fig. S2a for direct comparison with controls of similar age and Fig. S3 for statistical analysis without longitudinal samples. With the onset of acute neuroinflammatory demyelination (CALD), NfL levels were markedly increased to levels by far exceeding those observed in AMN and controls (model estimate of mean ratio vs. AMN: 11.0; adj. 95% CI: 6.2 – 19.7; adj. p<0.001; estimate of ratio vs. controls: 21.4; adj. 95% CI: 13.7 – 33.5; adj. p<0.001; Fig. 1a). See Fig. S2b for display of the data and statistical analysis upon separation of the CALD group into childhood/adolescence and adulthood onset.

Aging (Fig. 1c) affected NfL in healthy controls (0.24 pg/ml increase per year) and asymptomatic X-ALD/AMN patients (0.14 pg/ml increase per year) to a similar extent ($p=0.109$). In CALD patients presenting with highly elevated blood NfL, extensive axonal destruction masked age-related effects on blood NfL quantities (Fig. 1d).

NfL strongly associates with MRI scored CALD brain lesion severity

With onset of CALD, blood NfL levels were markedly increased in the majority of affected X-ALD patients (Fig. 1a, d). To further understand the extent of NfL amplification in individual CALD patients, we performed a linear regression analysis to assess the relationship between NfL and the activity of the inflammatory myelin destruction (Loes score of the MRI brain lesions). We found a statistically significant dependence of $\log(\text{NfL})$ on the status of white matter abnormalities ($R^2=0.73$, $p=0.002$), with more advanced CALD patients presenting both higher NfL and higher Loes scores (Fig. 2). There was no difference between the CALD cohort with childhood/adolescent onset and patients developing CALD in adulthood. In CALD patients with very high Loes scores of ≥ 17 , NfL levels were lower than in patients with Loes scores of 16, possibly due to more extensive advanced lesions with complete loss of large-calibre axons. In two boys and one adult patient with CALD identified at a very early disease stage (Loes MRI severity score of 1 and 2), NfL levels were still higher than in controls of similar age (Fig. 1d, Fig. 2).

Figures

Figure 1 – NEW FIG. 1 d

Fig. 1. NfL levels in X-ALD patients. **a** NfL in plasma and serum samples of asymptomatic X-ALD patients (n=7, median age=31 years, total sample number=8), non-inflammatory AMN (n=61, median age=40 years, total sample number=93), inflammatory CALD (n=24, 13 childhood/adolescent CALD cases and 11 adult CALD cases, median age=17 years, total sample number=32), and healthy controls (n=55, median age=37 years, total sample number=56). The median NfL level is indicated by a horizontal line. Total sample numbers include samples collected longitudinally from the same healthy control or patients during disease progression. Comparison of log(NfL) levels was done using a linear mixed model adjusted for sample type (serum vs. plasma) with addition of a random ID factor to account for longitudinal sampling of some individuals. Multiple testing was corrected by Tukey's method. Adjustment of the group comparisons for age differences did not change the significances or the reported p-values. **b** NfL in adult X-ALD patients with AMN status subdivided into: AMN patients without brain involvement at time of sampling (nonconverting AMN, n=41, median age=42 years), AMN patients with self-arrested brain lesions (CALD-arrested, n=10, median age=39 years), and AMN patients that at the time of blood sampling were free of any inflammatory activity but later (median duration until diagnosed conversion=3.5 years post sampling) developed CALD (premanifest-CALD, n=10, median age=40 years). In cases of sampling at several time points during disease progression, the latest sample was used for display and analysis. The potential of NfL for discriminating AMN from premanifest CALD measurements is investigated in a logistic regression model. **c** Linear regression of NfL on age in samples from X-ALD patients with asymptomatic/AMN status and adult healthy controls. Data points in orange (premanifest-CALD) indicate patients that were pre-symptomatic at baseline but converted to CALD later. Data points in green indicate asymptomatic X-ALD patients. **d** Association of NfL and age in samples from CALD patients and healthy controls. For longitudinal samples, the latest time point is marked in black and preceding ones are indicated in blue.

Discussion, p.11

Thirdly, for our findings to be generally applicable to the clinical management of CALD patients, with NfL levels supporting decision making and clinical interventions, more detailed investigation of the predictive power of NfL in individual CALD patients will be needed, with consideration also of a smoldering disease course and enlarged age-matched control groups for childhood/adolescent CALD patients.

3. Furthermore, while I understand that reference levels and cut-offs are to premature to suggest, I believe that the levels seen in this paper should be elaborated more on in comparison to other diseases (you briefly mention MS).

To facilitate the direct comparison of NfL levels in X-ALD patients with those in other disorders, we have included values reported for other neurodegenerative disorders (MS and ALS) in the Discussion section.

Changes in the manuscript:

Discussion, p.9

In a comparatively large set of X-ALD samples, we found that blood NfL is moderately but significantly elevated in AMN patients, reaching similar levels as observed in multiple sclerosis in remission (mean NfL, 17.0 pg/ml)²⁵, and strongly amplified with the onset of acute inflammatory brain demyelination in X-ALD patients, here aligning more with the levels reported in amyotrophic lateral sclerosis (median NfL, 125 pg/ml)²⁹.

Specific issues:

4. Figure 3: While the FLAIR is a T2W signal I think it should be mentioned that Fig 3 shows FLAIR sequences.

We thank the reviewer for this comment and have changed the figure legend of Fig. 3 accordingly.

Changes in the manuscript:

Figure legends:

Fig. 3. MRI analysis of slowly progressive smoldering inflammation in CALD. a T2-weighted (FLAIR) sequence before onset of CALD in patient CALD4 with classical, rapidly progressive disease course. **b, c** Frontal lobe cerebral inflammatory lesion with pronounced Gd-enhancement in T1-weighted (T1W) images and elevated FLAIR signal in the genu of the corpus callosum and frontal white matter, which was not seen two years earlier (a) and with severe progression of the enhancing lesion seven months later. **d,e** MRI scans of a CALD patient (CALD2) with atypical, slowly progressive smoldering inflammation. **d** Hazy Gd-enhancement in T1W images and mildly elevated FLAIR signal in the splenium of corpus callosum and posterior white matter. **e** Four years later, lesions had progressed mildly with diffuse marginal Gd-enhancement still notable, suggesting chronic smoldering inflammation.

5. Please use only 3 significant numbers (e.g. $p < 0.001$ and not 0.0012). Your use of 3 or 4 varies throughout the manuscript.

We have now consistently changed the display of p values to 3 significant digits.

Specific comments to referee 2:

Reviewer #2 (Remarks to the Author):

1-In my view, the EDSS is a good qualitative scale to assess the general status of patients with AMN, however I think that it is not a scale sensitive enough to be able to detect small changes in patients. It would therefore be advisable to correlate NfL levels to more quantitative scales such as 6mWT or 2mWT in these patients as an outcome measure, or to the MRI lesions in cervical spinal cord as discussed; perhaps a more in-depth discussion on the limitations of EDSS is required here.

Due to the low progression rate, we indeed agree with referee 2 on shortcomings of the EDSS rating system, which was originally developed for patients with MS, for overall evaluation of NfL as a biomarker in the context of the clinical status of AMN. Accordingly, we have now additionally correlated NfL levels in AMN patients to the motor function grading system of the Adult Adrenoleukodystrophy Clinical Score (AACS), which is a disease-specific scoring system that has been specifically developed for the assessment of clinical phenotypes in adult X-ALD patients (Köhler et al., A new disease-specific scoring system for adult phenotypes of X-linked adrenoleukodystrophy, *J Mol Neurosci* 13, 247-252 (1999)). We found a moderate correlation of NfL with AACS-graded myelopathy in AMN patients ($r_{\text{Spearman}}=0.376$), but in contrast to EDSS, the average increase of 4.6% in NfL levels with each additional AACS-motor function grading point did not reach statistical significance ($p=0.246$). We have incorporated this data in new Supplementary Figure S7 and added details on the AACS score as new Supplementary Table S4. We additionally correlated the NfL values of AMN patients also to the complete AACS system that, by incorporating next to motor function also bladder, sensory and cerebral involvement, is intended to differentiate between adult X-ALD phenotypes. As expected from this AMN sample set, with the majority of samples derived from “pure” AMN patients lacking cerebral involvement, the overall increase in NfL levels with each additional AACS grading point was reduced to 1.1% ($p=0.636$).

Of note, during revision of this manuscript, a report by Ballegoij et al. that focused on NfL in AMN patients but with shorter longitudinal sampling time compared to our study appeared in *Annals of Clinical and Translational Neurology* (October 2020). In this publication, the authors demonstrated that clinical parameters of myelopathy scored either by EDSS, SSPROM or timed up-and-go were associated with blood NfL. We have included a statement to this observation in the discussion section and incorporated the citation of this recent publication within our manuscript.

Changes in the manuscript:

Results, p. 8

With EDSS being a neurological disability scoring system originally developed for patients with multiple sclerosis, we additionally evaluated the neurologic dysfunction in AMN patients by correlating NfL levels to the motor function assessment based on the Adult Adrenoleukodystrophy Clinical Score (AACS). The AACS is a disease-specific scoring system that has been generated to differentiate clinical phenotypes in adult X-ALD²⁹, details on the AACS gradings are shown in Table S4. Whereas the analysis revealed a moderate correlation of NfL with AACS-graded motoric dysfunction in AMN patients ($r_{\text{Spearman}}=0.376$), the average increase of 4.6% in NfL levels with each additional AACS-motoric function grading point did not reach statistical significance ($p=0.246$; Fig. S7).

Figures: NEW Suppl. Fig. S7

Fig. S7. Worsening of AACS-graded myelopathy in AMN patients is not significantly reflected by increased NfL. The potential dependence of log of NfL on AACS graded (a) motoric and (b) combined motor, bladder, sensory and cerebral functions in individual asymptomatic X-ALD and AMN patients (n=60, total sample number including longitudinal samples=93) was analysed using a linear mixed model with fixed effects AACS-motoric function grading or combined AACS grading as well as age and sample type as adjustment variables and a random ID factor. (a) Average increase of 4.6% in NfL levels with each additional AACS-motoric function grading point, $p=0.246$; (b) Average increase of 1.1% in NfL levels with each additional combined AACS grading point, $p=0.636$. AACS, Adulthood ALD/AMN Clinical Scoring System.

Discussion, p. 10 and 11

In AMN, the minor increase in NfL levels with EDSS-graded worsening of the myeloneuropathy and the moderate correlation with both EDSS- and AACS-graded motoric functions ($r_{\text{Spearman}}=0.354$ and $r_{\text{Spearman}}=0.376$, respectively) are not entirely surprising, since AMN is characterized by a slow progression rate based on traditional outcome measures like EDSS.

During revision of this manuscript, van Ballegoij and colleagues reported increased levels of NfL in both male and female AMN patients and found that in male patients, blood NfL was associated with clinical parameters of myelopathy as scored by EDSS, Severity Scoring system for Progressive Myelopathy (SSPROM) and timed up-and-go³⁸. Accordingly, these investigations further support our observation that NfL may be of value also in the context of AMN.

2-Is not clear to this reviewer why there is no control samples for the CALD children; the levels of NfL should be quantified also in control children.

We thank the reviewer for this valid point that enables a direct comparison of NfL levels in boys with CALD and controls of similar age. We were able to obtain samples from seven healthy control children and have included the data in Fig. 1a, new Fig. 1d and new Suppl. Fig. S2. This aspect was also treated in the response to reviewer 1, point #3.

Changes in the manuscript:

cf. changes listed in response to reviewer 1 #3.

3-The legends of Figure 1, Figure 2, and Figure 6 do not include the statistical analysis used, needs to be corrected.

We have now included details on the statistical analysis used in Fig. 1, Fig. 2 and Fig. 6.

Changes in the manuscript:

Figure Legends:

Fig. 1. NfL levels in X-ALD patients. **a** NfL in plasma and serum samples of asymptomatic X-ALD patients (n=7, median age=31 years, total sample number=8), non-inflammatory AMN (n=61, median age=40 years, total sample number=93), inflammatory CALD (n=24, 13 childhood/adolescent CALD cases and 11 adult CALD cases, median age=17 years, total sample number=32), and healthy controls (n=55, median age=37 years, total sample number=56). The median NfL level is indicated by a horizontal line. Total sample numbers include samples collected longitudinally from the same healthy control or patients during disease progression. Comparison of log(NfL) levels was done using a linear mixed model adjusted for sample type (serum vs. plasma) with addition of a random ID factor to account for longitudinal sampling of some individuals. Multiple testing was corrected by Tukey's method. Adjustment of the group comparisons for age differences did not change the significances or the reported p-values. **b** NfL in adult X-ALD patients with AMN status subdivided into: AMN patients without brain involvement at time of sampling (nonconverting AMN, n=41, median age=42 years), AMN patients with self-arrested brain lesions (CALD-arrested, n=10, median age=39 years), and AMN patients that at the time of blood sampling were free of any inflammatory activity but later (median duration until diagnosed conversion=3.5 years post sampling) developed CALD (premanifest-CALD, n=10, median age=40 years). In cases of sampling at several time points during disease progression, the latest sample was used for display and analysis. The potential of NfL for discriminating AMN from premanifest CALD measurements is investigated in a logistic regression model. **c** Linear regression of NfL on age in samples from X-ALD patients with asymptomatic/AMN status and adult healthy controls. Data points in orange (premanifest-CALD) indicate patients that were pre-symptomatic at baseline but converted to CALD later. Data points in green indicate asymptomatic X-ALD patients. **d** Association of NfL and age in samples from CALD patients and healthy controls. For longitudinal samples, the latest time point is marked in black and preceding ones are indicated in blue.

Fig. 2. Association between plasma/serum NfL and MRI score of brain lesions in CALD patients. The MRI score of location and activity of the inflammatory myelin destruction in CALD patients (n=24, including 13 childhood/adolescent CALD cases and 11 adult CALD cases, median age=17 years, total sample number=32), according to the 34-point severity scale of Loes, was associated with NfL levels ($R^2= 0.73$, $p=0.002$). Arrowheads indicate three patients with atypically mild and slowly progressive adult CALD disease course (CALD1-3). Statistical analysis included a fixed factor for sample type and a random ID factor as well as a linear and quadratic term for the Loes score.

Fig. 6. Worsening of EDSS-graded myelopathy in AMN is reflected by increased NfL. The potential dependence of log of NfL on EDSS graded clinical severity in individual asymptomatic X-ALD and AMN patients (n=68, total sample number=101 samples) was analysed using a linear mixed model with fixed effects EDSS as well as age and sample type as adjustment variables and a random ID factor. The adjusted slope of the linear regression is back-transformed to the original scale, thus resulting in a percentage change, estimated as 6% higher NfL for each additional EDSS score point (95% CI 0.7% to 11.4%, $p=0.026$).

4-In Figure 1A, the child/adolescent and adult CALD data are put together; I would very much prefer to see the results when the two groups are separated. Also, I do not understand why several samples of the same patient are used. In AMN, there is n=61 patients, but 93 samples; in asymptomatic X-ALD there are n=7 patients but 8 samples; in CALD there are n=24 patients but 32 samples; and there are n=48 healthy individuals but 49 samples...Are those replicates of the same sample? How would the analysis look like without replications from the same sample?

We agree with the reviewer that a separate analysis of both, childhood/adolescent CALD and adult CALD, with corresponding healthy control groups would add informative value to the manuscript. Accordingly, we have obtained samples from age-matched childhood/adolescent controls, split the groups, reanalysed the data and added the result as new Suppl. Fig. 2.

In Fig. 1a, also longitudinal samples from patients throughout disease progression and from one healthy control at two different time points are included. Consequently, these samples are not technical replicates but represent individual data points for NfL at a certain disease stage or, for one healthy control, at a later point in time. In order to avoid misunderstanding and to further clarify this aspect, we have modified the figure legend accordingly. In addition, we present the data without longitudinal samples as new Suppl. Fig. 3.

Changes in the manuscript:

Results, p. 5

With the onset of acute neuroinflammatory demyelination (CALD), NfL levels were markedly increased to levels by far exceeding those observed in AMN and controls (model estimate of mean difference: 3.09; adj. 95% CI: 2.13 – 4.04; $p<0.001$; Fig. 1A). See Fig. S2b for display of the data and statistical analysis upon separation of the CALD group into childhood/adolescence and adulthood onset.

Figures: NEW Suppl. Fig. 2

Fig. S2. Comparison of NfL levels in X-ALD patients and healthy controls of similar age.

(A) NfL in plasma and serum samples of non-inflammatory AMN patients (n=61, median age=40 years, total sample number =93) and healthy adult controls (n=48, median age=39 years, total sample number=49) (B) NfL in plasma and serum samples of inflammatory childhood/adolescent CALD (n=13, median age=12, total sample number=19), adult CALD patients (n=11, median age=44, total sample number=13) and healthy controls of similar age (adults: n=48, median age=39, total sample number=49; childhood/adolescent: n=7, median age=11, total sample number=7). The median level is indicated by a horizontal line. Total sample numbers include samples obtained longitudinally from the same individuals (healthy controls or individual patients throughout disease progression). Longitudinal samples are indicated in blue, the latest time point of sampling is marked in grey (healthy controls) and black (X-ALD). Comparison of log(NfL) levels was done using a linear mixed model adjusted for sample type with addition of a random ID factor to account for longitudinal sampling of some individuals. In (B), the magnitude of difference between CALD and controls is significantly different between children/adolescents and adults ($p=0.005$). However, both comparisons within the two age groups are highly significant (both $p<0.001$).

NEW Suppl. Fig. 3

Fig. S3. NfL levels in X-ALD patients without longitudinal sampling. NfL in plasma and serum samples of asymptomatic X-ALD patients (n=7, median age=26), non-inflammatory AMN (n=61, median age=42), inflammatory CALD (n=24, median age=19) and healthy controls (n=55, median age=37). Values represent the first time point of sampling. The median NfL level is indicated by a horizontal line. Comparison of log(NfL) levels was done using a linear mixed model adjusted for sample type (serum vs. plasma). Multiple testing was corrected by Tukey's method.

5-In Figure 1B, AMN patients are divided into AMN, CALD arrested, and premanifest CALD. Why did the authors use only one replicate here? The mean that we are seeing in the figure, is the mean of the replicate?

For some AMN patients we were able to obtain longitudinal samples representing different time points during disease progression. Accordingly, these samples are not technical replicates but represent individual data points for NfL at a certain disease stage. In Fig. 1b, some AMN patients are included for which such longitudinal measurements are available. However, to enable a more appropriate comparison, especially between the AMN and pre-manifest CALD groups, we here included only the samples from the latest time point of disease progression for the individual patient. To further clarify and avoid confusion on this issue, we have changed the figure legend accordingly.

Changes in the manuscript:

cf. changes listed to Fig. 1 legend above in response to reviewer 2 #3.

6-Figure 4B and 5 shows CALD4, CALD8, and CALD9 patients who developed classical CALD with highly active lesions and exhibited boosted levels of NfL associated with the presence of Gd-enhancement. The authors have plasma of these 3 patients with the two different phenotypes (AMN and CALD). Have the authors tried to do a statistical analysis of NfL levels after conversion of AMN into CALD?

We understand the suggestion of reviewer 2 for performing statistical analysis on the data derived from three patients. However, due to the low sample size of three, there is a high chance of a false negative finding here. Therefore, we prefer to leave it with the descriptive approach for this manuscript.

7-In Figure 6, the authors show that levels of NfL correlate to an increase of EDSS, chosen as a relevant outcome measure. They used all the replicates (101 samples but n=68 patients) from asymptomatic X-ALD and from AMN patients, as in Figure 1A. In Figure 1B, the authors observe that levels of NfL were significantly higher in AMN patients who converted to CALD. What would the results be regarding correlation to EDSS if these samples (AMN patients who developed classical CALD) are removed for the analysis? Would the correlation between EDSS and NfL levels hold true?

In Figure 6, our intention was to include all X-ALD patients, either still asymptomatic or already presenting clear signs of myelopathy but lacking any signs of neuroinflammation, as

these patients would normally be scored by clinicians during routine investigations as “AMN”. However, we agree that it would be of interest whether an ex-post exclusion of AMN patients that later on developed CALD would change the correlation between EDSS and NfL. Thus, with the limitation that patients might be still included in the analysis who later will convert to CALD, we excluded the already known “premanifest” CALD patients and performed statistics. With a p-value of 0.053, the correlation analysis revealed borderline significance between EDSS and NfL. We have included this data as new Suppl. Fig. S6.

Changes in the manuscript:

Results, p.8

We found that worsening of the myeloneuropathy in AMN patients is reflected by increased NfL levels, with each additional EDSS score point resulting in an average 6% increase in NfL (95% CI 0.7% to 11.4%, $p=0.026$), when adjusting for age at measurement and sample type (Fig. 6, cf. Fig. S6 for ex-post exclusion of AMN patients that later developed CALD (“premanifest CALD”)).

Figure: New Suppl. Fig. 6

Fig. S6. Association of EDSS-graded myelopathy and blood NfL levels in AMN patients upon ex-post exclusion of AMN patients that later on developed CALD (“premanifest CALD”). The potential dependence of log of NfL on EDSS-graded clinical severity in individual asymptomatic X-ALD and AMN patients ($n=58$, total sample number= 80) with exclusion of patients that developed CALD (“premanifest CALD”) during the investigational time of this study was analysed using a linear mixed model with fixed effects EDSS ($p=0.053$) and with age and sample type as adjustment variables. EDSS, Expanded Disability Status Scale.

Reviewer #1 (Remarks to the Author):

I applaud the authors for their comments and revisions. The context of their findings are much better described in these revisions. I have no additional comments and hereby endorse the manuscript for publication.

Reviewer #2 (Remarks to the Author):

The authors have satisfactorily addressed the requests.